# Hypomineralized Primary Teeth in Preterm Low Birth Weight Children and Its Association with Molar Incisor Hypomineralization—A 3-Year-Prospective Study

**DOI:** 10.3390/children8121111

**Published:** 2021-12-02

**Authors:** Roshan Noor Mohamed, Sakeenabi Basha, Shruti Girish Virupaxi, Neena Idawara Eregowda, Poornima Parameshwarappa

**Affiliations:** 1Department of Pedodontics, Faculty of Dentistry, Taif University, P.O. Box 11099, Taif 21944, Saudi Arabia; roshan.noor@tudent.edu.sa; 2Department of Community Dentistry, Faculty of Dentistry, Taif University, P.O. Box 11099, Taif 21944, Saudi Arabia; 3Department of Pediatric Dentistry, Maratha Mandals NGH Institute of Dental Sciences, Belagavi 590001, India; dr.shrutigv@gmail.com; 4Department of Pedodontics and Preventive Dentistry, College of Dental Sciences, Davangere 577004, India; neena.ganesh@gmail.com (N.I.E.); drpoornimas@gmail.com (P.P.)

**Keywords:** enamel hypomineralization, molar incisor hypomineralization, preterm low birth weight, primary molars

## Abstract

The present prospective study was conducted to assess the prevalence of enamel hypomineralization (EH) in primary dentition among preterm low birth weight (PT-LBW) children, incidence of molar incisor hypomineralization (MIH) in the same cohorts, and to determine associations between PT-LBW, hypomineralization in primary second molars, and MIH. A total of 287 PTLBW study subjects and 290 control full-term normal birth weight subjects were followed up for 36 months. Enamel defects were recorded at baseline. The same cohorts were examined after 3 years for MIH using the European Academy of Paediatric Dentistry (EAPD) criteria. Multiple variable logistic regression models were developed. A total of 279 children (48.4%) presented with EH in primary dentition and 207 (35.9%) children presented with MIH. Children with primary second molar hypomineralization had 2.13 (R^2^ = 0.19, 95% CI = 0.98–4.19, *p* = 0.005) times higher frequency of MIH. Children with PT-LBW had 3.02 times (R^2^ = 0.31, 95% CI = 1.01–5.94, *p* = 0.005) higher frequency of MIH incidence after adjusting for childhood infection, prenatal history, and presence of hypomineralized primary second molars. To conclude, the present study showed significant association between PT-LBW, hypomineralized second primary molars, and incidence of MIH.

## 1. Introduction

Enamel hypomineralization (EH) in primary teeth and molar incisor hypomineralization (MIH) in permanent teeth are qualitative defects of enamel resulting from disturbances during the matrix formation stage of enamel development [1,2,3]. The etiologies of EH and MIH are considered multifactorial and may be influenced by systemic, genetic and or environmental factors, which include premature birth, underweight birth, infections, hypoxia, malnutrition, or metabolic disorders, and are often reported in higher frequency among low socioeconomic families [4,5,6,7,8,9]. These hypomineralized areas are responsible for considerable esthetic problems, hypersensitivity of involved teeth and predisposition to dental caries both in primary and permanent dentition, affecting children’s quality of life [10,11,12,13,14]. The reported rate of prevalence for EH ranges widely between 25–45.4% in primary dentition [2,4,10,15,16] and for MIH from 8.6% to 21.4% in permanent dentition [9,12,17,18] depending on the geographic population, teeth examined and the method used for diagnosis of these lesions. These hypomineralized defects of dental enamel serve as biological markers since alterations that occur during pregnancy remain permanently recorded on the tooth surface [19]. Recent systematic reviews show significant association between preterm low birth weight and MIH prevalence [20], and increased risk of EH in primary teeth [21]. Depending on which part of the tooth is affected, we can characterize these defects as being of intrauterine origin or not [20,21]. Preterm birth and low birth weight have been associated with enamel hypomineralization both in primary and permanent dentition, probably due to incomplete enamel mineralization or maturation or due to limited distribution of nutrients available for enamel formation [20,21]. Previously, few researchers have reported enamel hypomineralization in second primary molars (SPM), which can predict molar incisor hypomineralization (MIH) due to overlapping periods of mineralization between SPM and permanent first molars (PFMs) and incisors [22,23,24]. A recent systematic review showed significant association between EH in SPM and MIH; however, the authors reported high heterogeneity among the studies included in the meta-analysis [25]. The present prospective study was conducted with the following objectives: to assess the prevalence of EH in primary dentition among PT-LBW and incidence of MIH in the same cohorts, and to determine associations between PT-LBW, hypomineralization in primary second molars and MIH.

## 2. Materials and Methods

The ethical clearance statement: This study was conducted according to the guidelines of the Declaration of Helsinki and approved by the Institutional Review Board (or Ethics Committee) of College of Dental Sciences (Protocol number-CODS/IRB/25/2015/01) date 1 December 2015.

### 2.1. Study Population, Design and Sampling Procedure

A prospective study was planned. The sample size was determined based on results from a pilot study (anticipated population prevalence of 0.45, ά error of 5%, and power of the study at 80%), resulting in a sample size of 245 children, which was rounded to 300 in each group (300 PT-LBW and 300 controls) to counter the dropouts. Public and private maternity hospitals were approached for permission to review the hospital records regarding PT-LBW children born in their hospital. Among nineteen hospitals, seven hospitals (three public and four private) gave permission to gather information on PT-LBW babies born at their hospital. A total of 12,732 baby delivery records were analyzed out of which 722 were born with PT-LBT. The addresses and phone numbers of parents were obtained from hospital records and tried for communication. Four-hundred eleven parents were contacted and the rest had either moved from that place or were residents of a distant place that could not be reached by the investigator. The children were born between 1 January 2012 and 31 December 2013. The study objective was explained to the 411 parents identified from the hospital records for having PT-LBW child, among which 325 responded positively for voluntary participation with written consent for providing the details of data to be collected and oral examination of their children. A control group of 300 healthy born children (full-term and normal birth weight) was selected from the same hospital records after consulting the parents and confirming their agreement for participation in the study. All the children belonged to low natural fluoride content (<0.5 mg/L) drinking water areas. Ethical clearance was obtained by the institutional review board, (CODS-25/12/2015) before initiation of the study.

Questionnaire: A pretested questionnaire was utilized (Cronbach alpha α = 0.85) to collect the following information:Sociodemographic details: age, gender, parental education, family income.Prenatal history: mothers’ medical history, infection during pregnancy, medication history, vitamin D deficiency or hypocalcemia, gestational diabetes, hypertension, pre-eclampsia.Perinatal history: Type of delivery (vaginal or caesarean section), premature birth, birth weight, prolonged delivery (perinatal information taken from hospital records).Postnatal history: Childhood infection and illness (asthma, urinary tract infection, otitis media, chickenpox, respiratory tract infection, rubella, tonsillitis, high fever, allergies, epilepsy, renal failure, cardiac problems), antibiotic usage, breastfeeding history during the first four years of life.

### 2.2. Oral Examination

Baseline oral examination was conducted from June 2016 to August 2016. For all the selected children, an appointment was fixed for their dental examination under a portable light source during daylight at home. Dental examination was done by a single examiner and the type of enamel defect in primary dentition was recorded. The examiner was blinded regarding the birth information. The presence of enamel defects was checked using disposable mouth mirrors after cleaning and drying the teeth with sterile gauge. The criteria described by the modified DDE index proposed by FDI, 1992 was used to record enamel defects on the buccal surfaces of each tooth [26]. The coding used was as follows; Code 1—Demarcated opacities, Code 2—diffuse opacities, Code 3—demarcated and diffuse opacities, Code 4—hypoplasia and Code 5—hypoplasia and opacities. The tooth surfaces with doubtful or questionable defects less than 1 mm diameter were scored as normal. The examinations were calibrated to establish intra-examiner correlation by repeating 10% of the examinations one week after the initial examination. There was a significant correlation with Kappa value of 0.94, *p* < 0.05 for DDE.

### 2.3. Followup Examination

The same cohorts were examined after 3 years for MIH during June 2019 to August 2019. Diagnosis of MIH was done according to the European Academy of Paediatric Dentistry (EAPD) criteria [27]. Permanent first molars (PFMs) and permanent incisors were examined in a clean and wet condition for the presence of MIH. Lesions larger than 1 mm were recorded. The MIH was coded as follows: 0 = No defect, 1 = Demarcated opacity, 2 = Post-eruptive breakdown (PEB), 3 = Atypical restorations, 4 = Tooth loss due to MIH (permanent first molars extracted due to MIH). Children were considered to be affected by MIH if one or more PFMs were involved with or without permanent incisor involvement. The examinations were calibrated to achieve a significant intra-examiner correlation concerning the diagnostic criteria of MIH (Kappa value of 0.90, *p* < 0.05). Along with hypomineralization, dental caries status was recorded for all children both at baseline and follow-up examination using World Health Organization 2013 criteria [28].

### 2.4. Exclusion Criteria

Enamel defects on all teeth, teeth with dental fluorosis and enamel defects on permanent incisors without the involvement of PFMs were excluded from MIH.

### 2.5. Birth Weight and Birth Type

Birth weight was categorized into three groups: very low birth weight (VLBW < 1599 g), low birth weight (LBW < 2499 g) and normal birth weight (NBW ≥ 2500 g). The type of delivery was categorized into three groups: less than 34 weeks of gestation, less than 37 weeks of gestation and ≥37 weeks of gestation [29].

### 2.6. Socioeconomic Status

Modified Prasad classification [30] was used to assess the per capita family income: children were categorized into one of the three SES: upper class (per-capita income > 4700 Rupees per month), middle class (per-capita income 1200–4700 Rupees per month), lower class (per-capita income < 1200 Rupees per month).

### 2.7. Statistical Analysis

Differences in proportion were tested using Kruskal–Wallis H followed by Mann–Whitney U tests for inter group comparison, and chi-squared tests. Differences in means were tested using analysis of variance (ANOVA) followed by Tukey’s post hoc and independent sample t-tests as necessary. Multiple variable logistic regression models were developed for MIH with odds ratios (OR) and 95% confidence intervals (CI) using PTLBW as explanatory factor after controlling for childhood infection, prenatal history and hypomineralized primary second molar. To avoid bias, dropout data were not included in the statistical analysis. The analysis was performed using the Statistical Package for Social Science version 17 (IBM SPSS Statistics, IBM Corp. in Armonk, NY, USA). All statistical tests were two-sided and the significance level was set at *p* < 0.05.

## 3. Results

During the 3-year follow-up, 23 children (13 in PTLBW group, 10 in control group) dropped out of the study (3.8% dropout rate), mostly due to parental job mobility. At baseline examination, the children were in the age range of 3.5-years-old to 4.3-years-old, with a mean age of 3.8 (±0.8) years. Among 287 PT-LBT children (average gestation period 35.1 weeks and average birth weight of 1.65 kg) who remained in the study group, 163 were girls and 124 were boys. Among 290 who remained in the control group, (average gestation period 39.5 weeks and average birth weight of 2.84 kg), 160 were girls and 130 were boys. In the study group (*n* = 287), 37 (22 boys and 15 girls) children belonged to the VLBW category with an average birth weight of 1.41 kg and 250 children (102 boys and 148 girls) were under LBW category with an average birth weight of 1.89 kg. A total of 11 children amongst the study group were born prematurely <34 weeks of gestation and 212 children were born <37 weeks of gestation. Sixty-four children were born full term, but with low birth weight (Table 1).

Table 2 shows the distribution of children with EH in primary dentition and MIH. A total of 279 children (48.4%) presented with EH in primary dentition and 207 (35.9%) children presented with MIH. Children with VLBW and LBW presented with a high frequency of EH in primary teeth and MIH compared to normal birth weight children (Figure 1). Preterm born children (<37 weeks of gestation) presented with significantly higher frequency of EH in primary teeth and MIH than children who were born after 37 weeks of gestation. The children with positive history of prenatal and childhood infection had significantly higher frequency of EH and MIH. Subjects with hypomineralized primary second molars had significantly (*p* = 0.001) higher incidence of MIH (76.7%).

The numbers of teeth affected with EH and MIH are summarized in Table 3, Figure 2. A total of 334 (16.4%) primary molars and 152 (13.3%) permanent molars were affected with enamel hypomineralization and MIH, respectively, in children with preterm low birth weight.

Demarcated opacity was the most frequent type of hypomineralization in primary teeth and primary second molars were the most frequently affected teeth (Table 4, Figure 3).

Table 5 and Figure 4 show the MIH category in permanent molars and incisors. Demarcated opacity was the most common type of MIH and maxillary molars were the most frequently affected teeth, followed by maxillary incisors, mandibular molars, and mandibular incisors.

Children with primary second molar hypomineralization had 2.13 (R^2^ = 0.19, 95% CI = 0.98–4.19, *p* = 0.005) times higher frequency of MIH. Children with PT-LBW had 3.02 times (R^2^ = 0.31, 95% CI = 1.01–5.94, *p* = 0.005) higher frequency of MIH incidence after adjusting for childhood infection, prenatal history, and presence of hypomineralized primary second molar (Table 6).

## 4. Discussion

The presence of enamel hypomineralization resulting from disturbances during matrix formation stage of enamel development increases the risk of dental caries both in primary and permanent dentition, thereby affecting a child’s quality of life [2,9,10,11,12,13,14]. These defects result from systemic, genetic or environmental factors and recent systematic reviews highlighted the influence of PTLBW on these enamel defects [2,3,4,5,6,7,8,9,20]. The prevalence of PTLBW is high among the Indian population because of low nutritional status [31,32], yet there is a lack of prospective studies at the population level to estimate the influence of PT-LBW with the prevalence of EH in primary teeth and MIH. The present prospective study was conducted to determine the association between hypomineralization in primary teeth among PTLBW children and MIH. The overall prevalence of EH in primary dentition (48.4%) and MIH incidence (35.9%) was slightly higher than previously published studies across the world [2,4,9,10,12,15,16,17]. This variation in prevalence may be attributed to study population involved, teeth examined, and the criteria for diagnosis.

The current study is the first to present data on the association between EH in primary teeth in PTLBW children and MIH. Children with PT-LBW had 3.02 times higher frequency of MIH prevalence after adjusting for prenatal, postnatal history, and primary teeth hypomineralization. A recent meta-analysis by Wu et al. [20] showed an odds ratio of 1.57 (95% CI 1.07–2.31) for MIH prevalence and preterm birth, and 3.25 (95% CI 2.28–4.62) for MIH prevalence and low birth weight. Incomplete enamel mineralization or maturation or due to limited distribution of nutrients available for enamel formation increases the prevalence of MIH and EH in these subjects [20,21].

Second primary molar (SPM) hypomineralization was significantly associated with MIH prevalence with an odds ratio of 2.13. This may be due to overlapping periods of mineralization between SPM and permanent first molars (PFMs) and incisors [22,23,24]. The meta-analysis by Garot et al. [25] showed a strong association (ORs 4.66, 95% CI 2.11–10.26, *p* < 0.001) between primary molar hypomineralization and MIH prevalence.

The present study results showed prenatal maternal infection and postnatal childhood infection significantly associated with primary teeth hypomineralization and MIH. The results are in agreement with past published studies [4,5,7,9,33,34,35]. Maternal/childhood infections can have a disturbing effect on ameloblasts during the late secretory or early maturation stage either directly or indirectly through pH shifts, malnutrition, hypoxia, an increase in temperature, or hypocalcemia leading to hypomineralization [7,33,34,35].

Study limitation: Chances of recall bias while collecting information on the history of prenatal and childhood infection cannot be ruled out. However, the prospective study design minimized the influence of recall bias, so that the extrapolation of study results to a larger population was not affected.

## 5. Conclusions

The overall prevalence of EH in primary dentition was 48.4% and MIH incidence in permanent dentition was 35.9%. Children with PT-LBW had 3.02 times higher frequency of MIH prevalence after adjusting for prenatal, postnatal history, and primary teeth hypomineralization. Second primary molar (SPM) hypomineralization was significantly associated with MIH incidence.

## Figures and Tables

**Figure 1 children-08-01111-f001:**
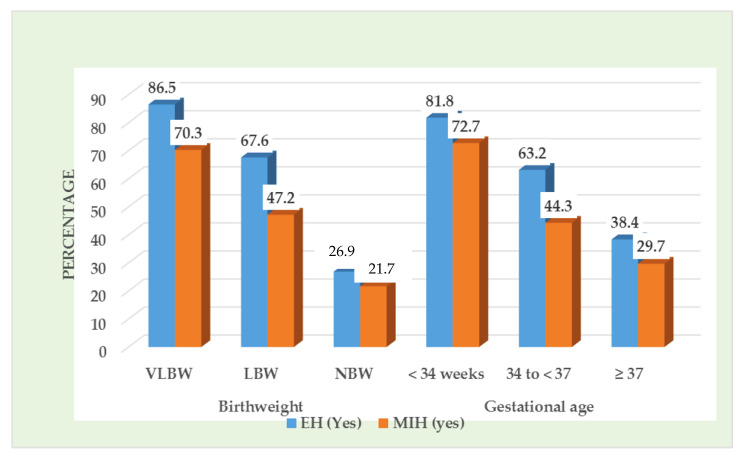
Presence of enamel hypomineralization (EH) and molar incisor hypomineralization (MIH) according to birthweight and gestational age.

**Figure 2 children-08-01111-f002:**
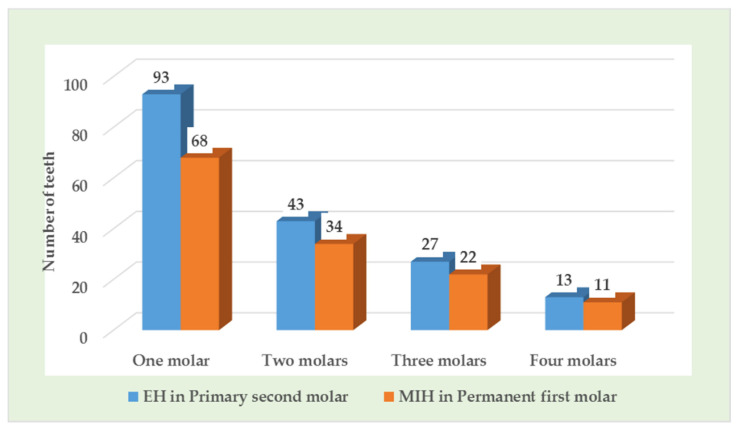
Number of primary second molars and permanent first molars affected per child with enamel hypomineralization (EH) and MIH.

**Figure 3 children-08-01111-f003:**
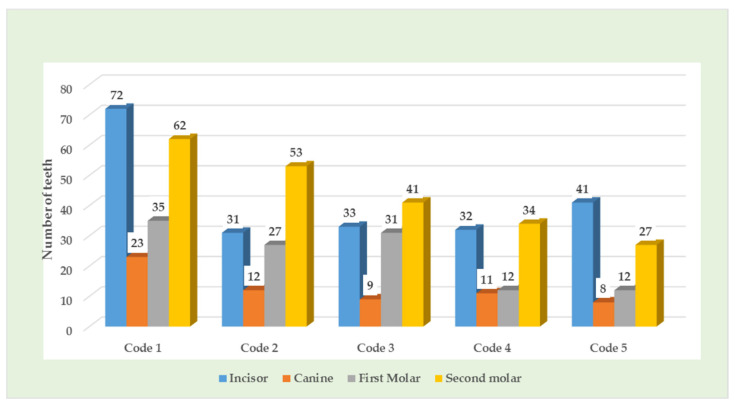
Number of primary teeth affected according to EH category in PTLBW children.

**Figure 4 children-08-01111-f004:**
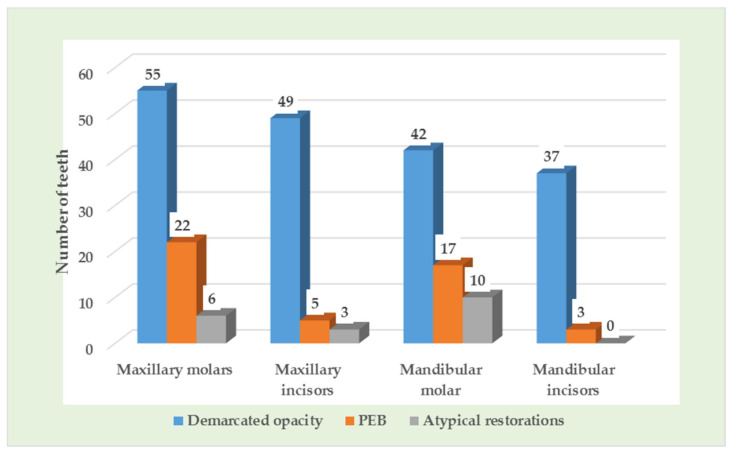
Number of permanent teeth affected according to MIH types.

**Table 1 children-08-01111-t001:** Descriptive characteristics of study participants.

Variables	Study Group *n* = 287	% or Range	Control Group *n* = 290	% or Range
**Gender**				
Boys	124	43.2	130	44.8
Girls	163	56.8	160	55.2
Chi-squared *p* value	0.12			
**Birth weight**				
VLBW	37	12.9	0	0
LBW	250	87.1	0	0
NBW	0	0	290	100
Mean birth weight in kg	1.65 ± 0.1	1050–1830 g	2.84 ± 1.2	2720–3900 g
**Gestational age**				
<34 weeks	11	3.8	0	0
34 weeks to <37 weeks	212	73.9	0	0
>37 weeks	64	22.3	290	100
Mean gestational age in weeks	35.1 ± 2.1	26–36	39.5 ± 2.2	37–42
**SES**				
Upper class	11	3.8	23	7.9
Middle class	113	39.4	98	33.8
Lowe class	163	56.8	169	58.3
Kruskal–Wallis *p* value	0.18			
**Prenatal history**				
Yes	123	42.9	63	21.7
No	164	57.1	227	78.3
Chi-squared *p* value	0.03			
**Postnatal History/Childhood Infection**				
Yes	135	47	179	61.7
No	152	53	111	38.3
Chi-squared *p* value	0.07			

VLBW—Very low birth weight (Birth weight of ≤1599 g), LBW—Low birth weight (Birth weight of >1599 g but ≤2499 g), NBW- Normal birth weight (NBW ≥ 2500 g), BMI—Body mass index, SES—socioeconomic status.

**Table 2 children-08-01111-t002:** Distribution of study participants with enamel hypomineralization in primary dentition and MIH incidence according to variables studied.

Variables	EH Yes (%)	EH No (%)	MIH Yes (%)	MIH No (%)
**Gender**				
Boys (*n* = 254)	111 (43.7)	143 (56.3)	92 (36.2)	162 (63.8)
Girls (*n* = 323)	168 (52.0)	155 (48.0)	115 (35.6)	208 (64.4)
Chi-Square, *p* value	0.135		0.126	
**Birth weight**				
VLBW ^a^ (*n* = 37)	32 (86.5)	5 (13.5)	26 (70.3)	11 (29.7)
LBW ^b^ (*n* = 250)	169 (67.6)	81 (32.4)	118 (47.2)	132 (52.8)
NBW ^c^ (*n* = 290)	78 (26.9)	212 (73.1)	63 (21.7)	227 (78.3)
Kruskal–Wallis, *p*	0.001		0.03	
Mann–Whitney U	a > b (U = 118, *p* = 0.031) a > c (U = 124, *p* = 0.001) b > c (U = 116, *p* = 0.022)		a > b (U = 98, *p* = 0.042) a > c (U = 112, *p* = 0.021) b > c (U = 102, *p* = 0.031)	
**Gestational age**				
<34 weeks ^d^ (*n* = 11)	9 (81.8)	2 (18.2)	8 (72.7)	3 (27.3)
34 to <37 ^e^ (*n* = 212)	134 (63.2)	78 (36.8)	94 (44.3)	118 (55.7)
≥37 ^f^ (*n* = 354)	136 (38.4)	218 (61.6)	105 (29.7)	249 (70.3)
Kruskal–Wallis, *p*	0.04		0.03	
Mann–Whitney U	d > f (U = 114, *p* = 0.036) e > f (U = 96, *p* = 0.042)		d > e (U = 92, *p* = 0.041) d > f (U = 112, *p* = 0.023) e > f (U = 94, *p* = 0.041)	
**SES**				
Upper (*n* = 34)	7 (20.6)	27 (79.4)	6 (17.6)	28 (82.4)
Middle (*n* = 211)	86 (40.8)	125 (59.2)	72 (34.1)	139 (65.9)
Lower (*n* = 332)	186 (56.0)	146 (44.0)	129 (38.9)	203 (61.1)
Kruskal–Wallis H, *p*	0.05		0.06	
**Prenatal history**				
Yes (*n* = 186)	57 (30.6)	129 (69.4)	42 (22.6)	144 (77.4)
No (*n* = 391)	222 (56.8)	169 (43.2)	165 (42.2)	226 (57.8)
Chi-squared, *p*	0.03		0.04	
**Postnatal History/Childhood Infection**				
Yes (*n* = 314)	202 (64.3)	112 (35.7)	142 (45.2)	172 (54.8)
No (*n* = 263)	77 (29.3)	186 (70.7)	65 (24.7)	198 (75.3)
Chi-squared, *p*	0.002		0.03	
**Subjects with Hypomineralized Primary Second Molars**				
Yes (*n* = 176)	NA		135 (76.7)	41 (23.3)
No (*n* = 401)	NA		72(18.0)	329 (82.0)
Chi-squared, *p*	0.001			

^a^ VLBW—very low birth weight, ^b^ LBW—low birth weight, ^c^ NBW—normal birth weight, ^d^—<34 weeks gestation, ^e^—34 to <37 weeks gestation, ^f^—≥37 weeks of gestation, SES—socioeconomic status, EH—Enamel Hypomineralization, MIH—Molar Incisor Hypomineralization.

**Table 3 children-08-01111-t003:** Number of primary teeth affected with hypomineralization and MIH in permanent dentition.

Study Group with Teeth	EH in Primary Teeth *n* (%)	MIH *n* (%)
PT-LBW		
Incisor	209/2061 (10.1)	97/2218 (4.4)
Canine	63/1011(6.2)	NA
Molar	334/2033 (16.4)	152/1139 (13.3)
**FTNBW**		
Incisor	79/2190 (3.6)	33/2308 (1.4)
Canine	20/1029 (1.9)	NA
Molar	144/2288 (6.3)	94/1153 (8.2)
**Number of SPM/PFMs Affected/Child**		
One molar	93 (52.8)	68 (50.3)
Two molars	43 (24.4)	34 (25.2)
Three molars	27 (15.3)	22 (16.3)
Four molars	13 (7.4)	11 (8.1)
Total	176	135
Mean (SD) affected molars	2.3 (1.1)	2.1 (0.9)

PTLBW—children with preterm and full-term low birth weight and very low birth weight, FTNBW—children with full-term normal birth weight, SPM—Second primary molar, PFMs—Permanent first molars, EH—Enamel Hypomineralization, MIH—Molar Incisor Hypomineralization.

**Table 4 children-08-01111-t004:** Number of primary teeth affected according to EH category.

Study Group with Teeth	Modified DDE Index by FDI, *n* (%)					
	Code 1	Code 2	Code 3	Code 4	Code 5	No DDE
**PTLBW**						
Incisor (*n* = 2061)	72 (3.5)	31 (1.5)	33 (1.6)	32 (1.5)	41 (2.0)	1852 (89.9)
Canine (*n* = 1011)	23 (2.3)	12 (1.2)	9 (0.9)	11 (1.1)	8 (0.8)	948 (93.7)
First molar (*n* = 924)	35 (3.8)	27 (2.9)	31 (3.4)	12 (1.3)	12 (1.3)	807 (87.3)
Second molar (*n* = 1109)	62 (5.6)	53 (4.8)	41 (3.7)	34 (3.1)	27 (2.4)	892 (80.4)
Kruskal–Wallis, *p* value	0.143					
**FTNBW**						
Incisor (*n* = 2190)	23 (1.0)	13 (0.6)	11 (0.5)	17 (0.8)	15 (0.7)	2111 (96.4)
Canine (*n* = 1029)	6 (0.6)	4 (0.4)	0	2 (0.2)	8 (0.8)	1009 (98.0)
First molar (*n* = 1138)	10 (0.9)	9 (0.8)	12 (1.0)	11 (1.0)	7 (0.6)	1089 (95.7)
Second molar (*n* = 1150)	19 (1.6)	25 (2.2)	17 (1.5)	16 (1.4)	18 (1.6)	1055 (91.7)
Kruskal–Wallis	0.162					

PTLBW—children with preterm and full-term low birth weight and very low birth weight, FTNBW—children with full-term normal birth weight, DDE—developmental defects of enamel, FDI—Federation Dentaire Internationale, Code 1—Demarcated opacities, Code 2—Diffuse opacities, Code 3—Demarcated and diffuse opacities, Code 4—Hypoplasia, Code 5—Hypoplasia and opacities.

**Table 5 children-08-01111-t005:** Number of permanent first molars and permanent incisors affected according to MIH types.

Study Group with Teeth	MIH Category According to EAPD, *n* (%)				
	Demarcated OPACITY	PEB	Atypical Restorations	Tooth Loss due to MIH	Tooth without MIH
**PTLBW**					
Maxillary molars (*n* = 568)	55 (9.7)	22 (3.9)	6 (1.0)	0	485 (85.4)
Maxillary incisors (*n* = 1106)	49 (4.4)	5 (0.5)	3 (0.3)	0	1049 (94.8)
Mandibular molar (*n* = 571)	42 (7.4)	17 (3.0)	10 (1.7)	0	502 (87.9)
Mandibular incisors (*n* = 1112)	37 (3.3)	3 (0.3)	0	0	1072 (96.4)
**FTNBW**					
Maxillary molars (*n* = 577)	42 (7.3)	4 (0.7)	2 (0.3)	0	529 (91.7)
Maxillary incisors (*n* = 1152)	18 (1.6)	0	0	0	1134 (98.4)
Mandibular molar (*n* = 576)	37 (6.4)	4 (0.7)	5 (0.9)	0	530 (9.2)
Mandibular incisors (*n* = 1156)	15 (1.3)	0	0	0	1141 (98.7)

PTLBW—children with preterm and full-term low birth weight and very low birth weight, FTNBW—children with full-term normal birth weight, MIH—Molar incisor hypomineralization, EAPD—European Academy of Paediatric Dentistry, PEB—Post eruptive breakdown.

**Table 6 children-08-01111-t006:** Multivariable logistic regression analysis—verification of effect of each confounding variables on the risk of MIH in preterm and low-birth weight children.

Variable	Odds Ratio (95% CI)	R^2^	*p* Value
PT-LBW	2.11 (0.97–4.08)	0.18	0.005
Subjects with primary second molar hypomineralization	2.13 (0.98–4.19)	0.19	0.005
Adjusted for childhood infection	2.32 (0.98–4.26)	0.22	0.005
Adjusted for childhood infection and prenatal history	2.37 (0.96–4.37)	0.24	0.005
Adjusted for childhood infection, prenatal history, and presence of hypomineralized second primary molar	3.02 (1.01–5.94)	0.31	0.005

PTLBW—Children with preterm and full-term low birth weight and very low birth weight.

## Data Availability

Data will be made available as per the request from corresponding author.

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
