# Peer review of "Hypomineralized Primary Teeth in Preterm Low Birth Weight Children and Its Association with Molar Incisor Hypomineralization—A 3-Year-Prospective Study"

_children, 2021, doi:10.3390/children8121111_

Round 1

Reviewer 1 Report

The article has a very current theme and is well written and structured. The methodology is very well defined with strict and well-explained criteria. However, authors need to make small changes to make the article more readable. The results presented in tables 2, 3, 4, and 5 must be placed in graphic format for better reading. The authors should also make a brief description of the results of these tables. In the discussion, the authors refer to the limitations of the study. However, they must clarify and expand on them. They should refer to any particular characteristics of the population studied and how this may or may not limit the extrapolation of their results to other populations. Authors should upgrade the bibliographic references, because the references they have are outdated, not having any references from 2021, for example.

Author Response

Response to Reviewer 1 Comments

Point 1: English language and style are fine/minor spell check required
Response 1: English language and grammar corrections were done using Grammarly soft ware

Point 2: The results presented in tables 2, 3, 4, and 5 must be placed in graphic format for better reading.

Response 2: The Figures were inserted as per the suggestion (Figure 1 to Figure 4 corresponding to Table 2 to Table 5).

Point 3: The authors should also make a brief description of the results of these tables.

Response 3: Brief description of the results of tables 2, 3, 4, and 5 is given as per the suggestion.

Point 4: In the discussion, the authors refer to the limitations of the study. However, they must clarify and expand on them. They should refer to any particular characteristics of the population studied and how this may or may not limit the extrapolation of their results to other populations.

Response 4: The limitation is expanded as per the suggestion

Point 5: Authors should upgrade the bibliographic references, because the references they have are outdated, not having any references from 2021, for example.

Response 5: The references were upgraded as per the suggestion with inclusion of research publications from 2021

Reviewer 2 Report

This is an interesting and extensive study on the association of enamel hypoplasias in primary dentition and MIH in preterm children with low and normal birth weight and the control group.

The study is well presented. There are several minor issues that should be adressed. 

Line 141 ''Low birth weight (LBW <2599 g) and normal birth weight (NBW ≥ 2500 g).'' The two categories overlap. Should the LBW be <2499g and not <2599 g?

Line 167 Please check mean birth weight in the control group. It is unlikely that it is 2.54 kg. Also the mean gestation period of 37.5 weeks seems low for the control group. ''Among 290 who remained in the control group, (average gestation period 37.5 weeks and average birth weight of 2.54 kg''

Table 1 line 176 As mentioned previously, mean weight in the control group could not be 2.54 kg if the recorded masses were in the range 2720-3900 g

Line 140 Criteria described here and in the legend of the Table 1 are not the same. Criteria for NBW, LBW and VLBW differ. Which was it?

Author Response

Response to Reviewer 2 Comments

Point 1: English language and style are fine/minor spell check required.

Response 1: English language and grammar corrections were done using Grammarly soft ware

Point 2: Line 141 ''Low birth weight (LBW <2599 g) and normal birth weight (NBW ≥ 2500 g).'' The two categories overlap. Should the LBW be <2499g and not <2599 g?

Response 2: LBW changed to <2499g as per suggestion.

Point 3: Line 167 Please check mean birth weight in the control group. It is unlikely that it is 2.54 kg. Also the mean gestation period of 37.5 weeks seems low for the control group. ''Among 290 who remained in the control group, (average gestation period 37.5 weeks and average birth weight of 2.54 kg''

Response 3: The typo errors in mean birth weight and gestational age in control group were corrected.

Point 4: Table 1 line 176 As mentioned previously, mean weight in the control group could not be 2.54 kg if the recorded masses were in the range 2720-3900 g

Response 4: The typo errors in mean birth weight and gestational age in control group were corrected.

Point 5: Line 140 Criteria described here and in the legend of the Table 1 are not the same. Criteria for NBW, LBW and VLBW differ. Which was it?

Response 5: Criteria for NBW, LBW and VLBW is corrected in legend of the table.